

# An Ensemble Observing System Simulation Experiment of Global Ocean Heat Content Variability

Arin D. Nelson[1], Jeffrey B. Weiss[1], Baylor Fox-Kemper[2], Royce K.P. Zia[3], and Fabienne Gaillard[4]

[1]Department of Atmospheric and Oceanic Sciences, University of Colorado, Boulder, Colorado, USA.
[2]Department of Earth, Environmental, and Planetary (DEEP) Sciences, Brown University, Providence, Rhode Island, USA.
[3]Center for Soft Matter and Biological Physics, Department of Physics, Virginia Polytechnical Institute and State University, Blacksburg, Virginia, USA.
[4]Ifremer, UMR 6523, LOPS, CNRS/Ifremer/IRD/UBO, CS 10070, Plouzane F-29280, France

*Correspondence to:* Arin D. Nelson (mr.adnelson@gmail.com)

**Abstract.** We quantify skill and uncertainty in observing the statistics of natural variability using observing system simulation experiments on an ensemble of climate simulations and an observing strategy of in situ measurements and objective mapping. The targeted statistic is the 0-700m global ocean heat content anomaly as observed by the In Situ Analysis System 2013 (ISAS13) strategy of a long, equilibriated simulation of the Community Climate System Model (CCSM) version 3.5. Sub-
annual variability is found to be significantly contaminated by the observing strategy, especially before 2005, primarily due to the sparseness and seasonality in the number and location of pre-Argo observations. However, one-year running means from 2005 onward are found to faithfully capture the natural variability of the model's true ocean heat content variability. During these years, synthetic observed annual running means are strongly correlated with the actual annual running means of the model, with a median correlation of 95%, versus only 60% for the observational record before 2005. When scaled to account
for the fact that the real ocean is more variable than the model, root mean square errors in observing the annual-running mean natural variability of the global ocean heat content are estimated to be 6.2 ZJ for the pre-Argo era (1990-2005) and 2.1 ZJ for the Argo era (2005-2013) with relative signal-to-noise ratios of 1.9 and 14.7. Combining the estimated, scaled uncertainties of the observing strategy with its estimated trend, the 1990-2013 trend in global ocean heat content is found to be $5.3 \pm 1.0$ ZJ/yr.

## 1 Introduction

The World Ocean is the largest active thermal reservoir in Earth's climate system, and therefore ocean heat content (OHC) variability plays a dominant role in Earth's energy balance (Levitus et al. (2001), Hansen et al. (2011), Trenberth et al. (2014)). Therefore, understanding the ocean's role in the Earth's climate system is of the utmost importance. At least since Levitus (1982), there have been many attempts to produce comprehensive global data collections of surface and subsurface ocean variables such as temperature and salinity. A diversity of these collections have been produced and differ in sources of observations and methods to produce global gridded data. Even with these differences, it is widely agreed that the heat content of the World
Ocean has increased over the last 50 years (Levitus (2000), Gouretski and Koltermann (2007), Domingues et al. (2008), Lyman and Johnson (2008), von Schuckmann and Le Traon (2011), Levitus et al. (2012), Roemmich et al. (2012), Balmaseda et al.



(2013), Hobbs and Willis (2013), Wunsch and Heimbach (2014)). However, the variability about the warming trend has yet to be reliably quantified due to the underlying uncertainties. Our focus here is to quantify the ability of an observing system to faithfully measure the variability about the warming trend by using an ensemble observing system simulation experiment (EOSSE).

There have been many attempts to quantify sources of error in observational ocean data products. They include uncertainties in the individual observations, uncertainties due to the non non-uniform spatio-temporal distribution of the observations, and uncertainties due to the global mapping method. Here we focus on the uncertainties due to the sampling and mapping methods, and summarize previous work in Section 2. These multiple sources of uncertainty make it difficult to estimate the uncertainties of observation-based ocean datasets, even for straightforward global variables such as ocean heat content. Fur-

thermore, reported uncertainties are subjective in their dependence on cross-comparison of different observational strategies of a single model simulation or reanalysis product. In other words, from these methods of estimating uncertainty, there is no way to quantify how the statistics of the observed ocean differ from those of the true ocean, what is quantified is how strategies differ relative to one another.

To more directly estimate the skill of an observing strategy, we propose a new EOSSE methodology: subject an ensemble of

independent global ocean model outputs to a single observing strategy and measure the distribution of 'observed' ocean states and the corresponding distribution of the modeled 'true' ocean states. Herein, metrics are recommended to be performed on the two distributions to quantify how 'observed' quantities differ from 'truth' as function of spatial and temporal scales, as well as by time and location. Methodological subjectivity persists in the choice of model and observing strategy, but the use of an EOSSE renders the method more powerful than either repeatedly sampling a single model output or inter-comparing different

observation-based datasets. This methodology is described in Section 3.

We demonstrate the method by quantifying the uncertainty of the observing strategy used in the creation of the In-Situ Analysis System 2013 (ISAS13) estimate of global ocean heat content (OHC) variability down to 700m between 1990 and 2013. The observing strategy will be applied to 37 independent model segments from the Community Climate System Model version 3.5 (CCSM3.5), from which the statistics of OHC variability will be compared between the 'observed' model segments

and their corresponding 'true' segments across a range of time scales. Here we focus on the uncertainty and skill due to the spatio-temporal distribution of sampling and the effects of the objective analysis, but this method can be easily used to investigate the effects on uncertainty and skill from other sources. The application of the methodology is described in Section 4, and the results are discussed in Section 5.

## 2   Previous Uncertainty Estimates

There is a substantial literature on the accuracy and precision of ocean instrumentation (e.g., Gouretski and Koltermann (2007), Abraham et al. (2013), Boyer et al. (2016)). Here, our focus is on the role of sampling and mapping methods, and we assume the observations are perfect. Errors in the individual ocean observations will render our estimates lower bounds on the true



uncertainties. Wunsch (2016) reviews methods to estimate the global ocean heat content, its trend, and the need for better uncertainty estimates.

One of the first studies to use an OSSE to study OHC was Gregory (2004), who compared the trend and variability of global OHC in the World Ocean Atlas 2001 (WOA01) with a HadCM3 model output subjected to the WOA01 observing

strategy. While they do not provide quantitative results, they found that both the HadCM3 output and the WOA01 had similar trends, and the latter had larger variability, especially outside of the well-observed northern hemisphere. However, they could not determine whether the difference in variability was due to the model having less variability than the ocean, the WOA observational over-estimating the variability, or some combination of the two.

AchutaRao et al. (2006), Lyman et al. (2006), and Lyman and Johnson (2008) investigated the bias and uncertainty in

estimating global heat content arising from the observing strategy used in the WOA. AchutaRao et al. (2006) performed synthetic observations of eight pairs of CMIP2+ coupled climate model runs (Meehl et al. (2000)). Each pair was from one of the CMIP2+ models, and one pair member was a control simulation with constant external forcing, while the other had $CO_2$ increasing at 1%/yr. They found that despite the sparseness of the data coverage, the observing strategy was capable of capturing the trend in global ocean heat content, but inflated estimates of variability in all ocean basins. In particular, they

found that the infilling procedure increased the variability of global ocean heat content over that from using observations alone. Lyman et al. (2006) and Lyman and Johnson (2008) used global sea-surface height fields to estimate synthetic ocean heat content anomalies to represent the true ocean and compared with measurements in the WOA; these studies focused on measuring the trends in ocean heat content rather than measuring the variability, which is the focus here.

As Argo became the dominant source of in-situ observations by the mid-2000's, studies of the spatio-temporal distribution

of observations began to focus specifically on the autonomous, free-drifting Argo floats. von Schuckmann and Le Traon (2011) found an OHC trend of $0.54 \pm 0.1$ W/m$^2$ for the time period of 2005-2010, and an uncertainty in the annual mean global upper ocean OHC (0-700m) values of $\pm 0.22 \times 10^8$ J/m$^2$ These uncertainties were computed using a box averaging method (Bretherton et al., 1976).

The study of Kamenkovich et al. (2009) is perhaps the most similar to the present one. They simulated the Argo observing

system (by sampling randomly with a specified number of locations or by simulating float trajectories) in an ocean general circulation model with increasing $CO_2$ forcing for years 1992 through 2001, which were then converted to global fields using the objective analysis procedure of Mariano and Brown (1992). The results of their OSSE quantify the errors of the reconstructed fields relative to their corresponding true fields from the full model and reveal causative factors. They estimate the error in the globally-averaged amplitude of the OHC annual cycle to be 9.25 W/m$^2$, and find the warming between annually averaged OHC

between years 1 and 10 to be 0.6 W/m$^2$ in their 'observations' compared to the 'true' model warming of 2.4 W/m$^2$. These tests were performed for various numbers of Argo floats in their advected and in random positions, which shows that biased float movement is the largest source of error in their simulated global OHC estimates. Increasing the number of floats significantly reduces these errors.

Another study to use OSSEs was Zhang et al. (2009), which simulated the 1976-2000 XBT observing system and the 2005

Argo observing system on one of the two fully-coupled general circulation models used in the Intergovernmental Panel on





Climate Change Fourth Assessment Report called CM2.0 (Randall et al. (2007)). One of their OSSEs compared the 1976-2000 complete observational network to the 2005 Argo network, both applied to 12 simulations with greenhouse gas forcing applied for the years coinciding to 1860 to 2000 in the simulations. Their results showed that the pre-Argo and 2005 Argo observations both resulted in root-mean-square (rms) errors in 0-700m global ocean temperature of $0.24\ ^o$C but had respective mean errors of $0.7 \times 10^{-2}\ ^o$C and $1 \times 10^{-2}\ ^o$C; the pre-Argo observations having a slightly smaller mean error.

von Schuckmann and Le Traon (2011) also looked at the sensitivity from using two different climatologies and found that the impact was small but not negligible, and was largely at yearly and smaller periods. Lyman and Johnson (2013) found that using a colder baseline climatology in a global mapping method produces larger warming trends. This was due to the fact that earlier time periods had much less global coverage than modern observations. Since observation-sparse regions tended towards zero anomaly in their mapping method, the larger the difference between the baseline climatology versus the actual climatology for the observation-rich Argo era, the larger the global trends observed.

Boyer et al. (2016) investigated the impact of different instrument bias corrections, baseline climatologies, and mapping methods. They found that for 1993-2008, the uncertainty due to instrument bias corrections varied between 10.9 to 22.4 ZJ, that due to mapping methods was 17.1 ZJ, and that due to baseline climatologies was 2.7 to 9.8 ZJ. They found the 1993-2008 trend was from 1.5 to 9.4 ZJ/yr depending on the choices. In all cases, their uncertainty results from the spread among various methods. Domingues et al. (2008), Levitus et al. (2012) and Good et al. (2013) estimated the uncertainty using a variety of statistical error propagation methods. Häkkinen et al. (2016) investigate the spatial structure of warming and interannual to multidecadal trends but did not quantify the variability or its uncertainty about these trends.

These previous works all use single model runs or multiple runs to look at sensitivity to various choices and to estimate uncertainty. Here, for the first time, we use ensemble techniques to measure the ability of an observing system to capture the true natural variability of a model and extrapolate those results to ocean observations.

## 3 Methodology

We propose to estimate the uncertainties of an observing strategy with the following procedure:

1. Gather an ensemble of independent[1] model simulations of the same temporal length as the observing strategy being tested.

2. Interpolate the model output from the model grid. to the locations and times of the observations to generate synthetic observations.

3. Apply the mapping method used in the observing strategy to the synthetic observations.

4. Calculate the desired statistics.

5. Compare the statistics between the 'observed' model and the 'true' model.

---

[1]Here the modeled intervals are sequential, non-overlapping windows of an equilibriated climate model described in Stevenson et al. (2012).





Many aspects of the 'observed' and 'true' oceans can be compared in the last two steps to attempt to quantify the skill of the observing strategy. Here the chosen foci are:

1. The cross-correlations of the 'observed' anomaly time series and the cross-correlations of the 'true' anomaly time series. Since the model ensemble members are independent, the 'true' time series are independent of each other. The cross-correlation of any two 'true' time series should thus approach zero. Some non-zero cross-correlation will exist, however, due to the finite length and density of the observational record. If the cross-correlations between pairs of 'observed' time series are statistically larger than the 'true' cross-correlations, then one can conclude that the observing strategy is contaminating the 'observed' estimates of the time series.

2. The correlations between corresponding 'observed' and 'true' time series generated from the same simulation ensemble member. If an observing strategy is perfect, the correlation should be unity. The closeness of these correlations to unity indicates how well the observing strategy estimates the 'true' time series.

Note that in (1) above the cross-correlations are generated for pairs of 'observed' ensemble members and pairs of 'true' ensemble members. In (2), the correlations are between pairs of 'observations' and 'truth' of the same ensemble member. Using these two quantifications of skill, one can choose among time scales that minimize (1) and maximize (2), which together define the time scales of variability that are the least contaminated by the observing strategy. These skill scores can also be applied to different subsets of the full observational record; for example, before and after the implementation of the Argo project. From the time scales one determines to be best observed by the observing strategy, the distribution of the differences between the 'observed' and 'true' time series can be used to quantify the bias and uncertainty due to the observing strategy.

## 4 The Observing Strategy

### 4.1 CORA Hydrographic Profiles

We used the quality-controlled global COriolis Ocean Dataset for Reanalysis (CORA) version 4.0 dataset of subsurface ocean temperature and salinity profiles from January 1990 through December 2013 Cabanes et al. (2013). It contains most hydrographic profiles available in this time period, including those from Argo; XBT, CTD, and XCTD casts from research and commercial vessels; moored and drifting buoys; and CTD attached to marine mammals (see Cabanes et al. (2013) for more details). All measurements with a quality control flag other than "reliable" were discarded. Observations beyond the nominal range of the current observations were also discarded, which include data poleward of $60^o$ North and South (Roemmich and Gilson (2009)) and beyond a depth of 700m (the maximum depth of most pre-Argo observations). From these retained observations, we use the date, latitude, longitude, and depths sampled to create synthetic observations.

Since the number (Figure 1) and distribution (Figure 2) of hydgrographic profiles in the CORA data set changes significantly from before to after the implementation of the Argo program, it is expected that the introduction of Argo improved global estimates of ocean variables. Therefore, along with testing the ISAS13 observing strategy for the entire time span of 1990 to 2013, a pre-Argo time span (1990-2005) and an Argo time span (2005-2013) are examined. The year 2005 is chosen as it is the





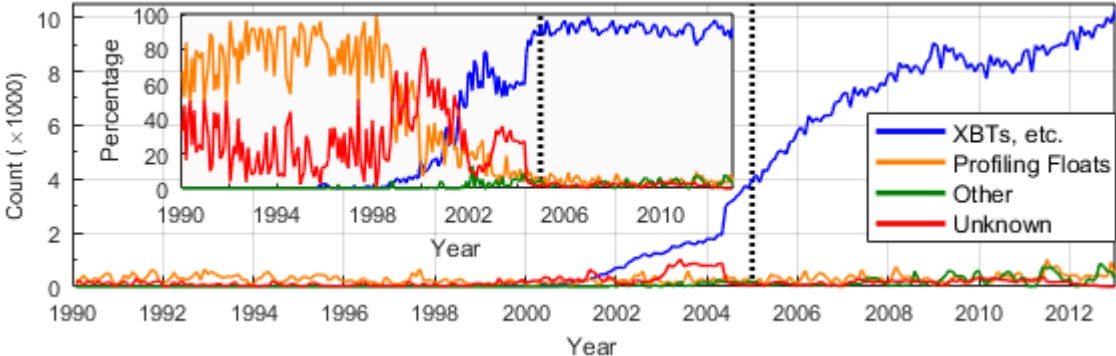

**Figure 1.** The number of retained hydrographic profiles per month. The black curve is the total number of profiles per month among all data types. The 'XBTs' curve includes XBTs, CTDs, and XCTDs. The 'Other' curve includes moored and drifting buoys, hydrocasts, thermistor chains, and sensors attached to marine mammals. Most 'Unknown' casts are from XBTs with an unlisted make and model, but no bias corrections relevant to XBT profiles are performed for these casts. The inset shows the same data but as a percentage of the total casts per month. The vertical dotted line indicates the cutoff between the pre-Argo and Argo eras.

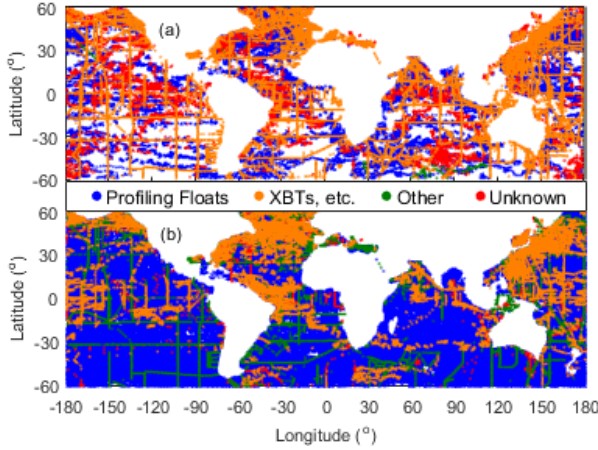

**Figure 2.** The locations of retained hydrographic profiles. Before Argo, observations were sparse and primarily by XBTs. Poor record keeping requires many of these observations to be from 'unknown' sources. The majority of the unsampled regions in the Argo era occur in regions shallower than 700m, although some are geopolitically sensitive areas.

date when the Argo program reached its intended level of global coverage, although Figure 1 indicates that observation density continued to climb until roughly 2009.



## 4.2 ISAS13 Observing Strategy

We use In-Situ Analysis System 2013 (ISAS13) observing strategy, which applies the In-Situ Analysis System tool verion 6 (ISAS-V6) on the CORA dataset by interpolating the CORA observations onto a standardized grid (Gaillard et al. (2016)). This objective analysis (OA) procedure uses the optimal interpolation technique presented by Bretherton et al. (1976) and depends on a mean reference (or first guess) field, a field of the expected variance about the mean field, and de-correlation length and time scales. The ISAS13 observing strategy employs this OA procedure using monthly means provided by the World Ocean Atlas 2005 (WOA05) monthly climatologies, monthly variances estimated from the observations relative to these climatologies, and decorrelation time scales chosen to represent 1) the resolution of the Argo network and 2) the first-mode Rossby radius of deformation on each point of their chosen analysis grid. Both decorrelation time scales were provided by the CORA dataset.

## 4.3 Global Ocean Heat Content Decomposition

The ocean heat content $q$ of some ocean grid box $i$ at time $t$ from the surface down to depth $z^*$ is estimated as follows:

$$q_i(t) = \rho c_p A_i \int_0^{z^*} \theta(t,z)dz, \tag{1}$$

where $q$ is in units of J, $\rho$ is seawater density taken to be 1025 kg m$^{-3}$, $c_p$ is the specific heat capacity of seawater taken to be 3850 J kg$^{-1}$ $^o$C$^{-1}$, $A_i$ is the surface area of gridbox $i$ in m$^2$, and $\theta$ is potential temperature in units of $^o$C. Seawater density and specific heat capacity are placed outside of the integral to simplify computations, which is acceptible since they change little in the global upper ocean [Warren, 1999]. Potential temperature for the CORA product is calculated from the given in-situ temperature and salinity profiles using the Thermodynamic Equation of Seawater 2010 (IOC and IAPSO (2010)).

The global ocean heat content down to depth $z^*$ at time $t$ is defined as follows:

$$Q_{z^*}(t) = \sum_i q_i(t), \tag{2}$$

where $i$ runs over all of the areas spanning the globe. Note that for our purposes, the region we define as "global" is limited to equatorward of $60^o$ N and S, and we chose $z^*$ to be 700m as stated earlier. For the remainder of this paper, the subscript $z^*$ will be dropped.

We decompose global ocean heat content into a long-time mean, a warming trend which we take to be linear, the long-term mean seasonal cycle, and a remaining anomaly:

$$Q(t) = Q_M + t\frac{\Delta Q}{\Delta t}(m) + Q_S(m) + Q_A(t). \tag{3}$$

Here $t$ is the time (in months) of the data, $m$ is the month-of-year of $t$, $\frac{\Delta Q}{\Delta t}(m)$ is the linear trend over the entire time period being considered (calculated separately for each month), $Q_S(m)$ is the fixed seasonal cycle defined by the all-time monthly



means, $Q_A(t)$ is the anomaly from the seasonal cycle and trend, and $Q_M$ is a constant such that the all-time mean of $Q_A(t)$ and $Q_S(m)$ are 0. This definition of $Q(t)$ is chosen so that $Q_A(t)$ encapsulates the "natural" variability of the ocean, defined here to be the variability of the ocean not forced by multi-decadal or slower linear climate change and seasonality. For simplicity, each month is given equal weight (i.e., they are approximated as having the same number of days). Also note that the linear

trends are computed separately for each month to ensure that $\frac{\Delta Q}{\Delta t}$ and $Q_S$ can be computed in any order, and that the mean of the 12 monthly linear trends is equivalent to the trend of the entire time series.

### 4.3.1 Temporal filtering

We expect that the skill of the ISAS13 observing strategy in capturing the natural variability will depend on the timescales of variability arising from both the actual dynamic variability of the ocean as well as the variability in both the location

and number of the instrumental casts. We expect that the variability due to the latter occurs primarily on higher frequencies. We thus decompose the anomaly time series $Q_A(t)$ into low-frequency and high-frequency components. The goal here is to choose a cut-off period which will contain the variability due to sampling in the high frequency component, thus produce a low frequency anomaly which will be representative of natural ocean variability. The decomposition is then

$$Q_A(t) = Q_L(t;\tau) + Q_H(t;\tau), \tag{4}$$

where $Q_L$ is the low frequency variability, $Q_H$ is the high-frequency variability, and $\tau$ is the cut-off period. We choose a simple filter: the low frequency component is defined in terms of a running mean with window size $\tau$, and the high-frequency component is the residual,

$$Q_L(t;\tau) = \int_{t-\tau/2}^{t+\tau/2} Q_A(t')dt',$$

$$Q_H(t;\tau) = Q_A(t) - Q_L(t;\tau). \tag{5}$$

### 4.4 CCSM 3.5 Model Output

We apply the ISAS13 observing strategy to an ensemble created using the National Center for Atmospheric Research (NCAR) Community Climate System Model (CCSM) version 3.5 in the fully coupled configuration (active atmosphere, ocean, land, and sea ice). The simulation is described in Stevenson et al. (2012) and is forced with steady 1990 concentrations of atmospheric carbon dioxide. The ocean model has a zonal resolution that varies from 340km at the equator to 40km around Greenland and 350km in the Northern Pacific. This spatially varying resolution is achieved by placing the north pole of the grid over

Greenland and reflects the different relevant length scales of the two processes that are important in maintaining a stable global climate; deep convection around Greenland and in the Arctic as well as ocean heat uptake at the equator. In the vertical there are 25 depth levels; the uppermost layer has a thickness of 8m and the deepest layer has a thickness of 500m. This and the atmospheric, land, and sea-ice models operate under the T31_3_3 setup, described in detail in Yeager et al. (2005), which have been developed specifically for long paleoclimate and biogeochemistry applications.





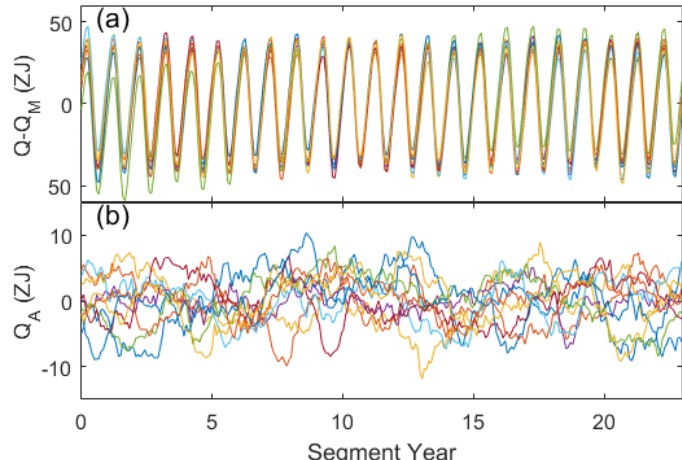

**Figure 3.** (a) The time series of (a) the global ocean heat content minus the segment mean, $Q - Q_M$, and (b) the anomaly $Q_A$, for 10 of the 37 23-year segments of the CCSM 3.5 output.

The coupled climate model used in this study was selected because it possesses subannual variability consistent with observations independent from those used here (e.g., Stevenson and Fox-Kemper (2010), Stevenson et al. (2012), Stevenson et al. (2013)), is run without variability in anthropogenic forcing, and it is long enough to capture even centennial variability. However, significantly more variability is presumed to be present in the real ocean, especially on the mesoscale spatio-temporal

range (100km, months) and below. Furthermore, real atmospheric synoptic variability is even more vigorous than is permitted in such a coupled model (Small et al., 2014). Thus, the estimates found using this model are based on a necessarily less variable climate system. In this sense, the estimates of sampling of variability here are a lower bound on the degree to which the real ocean variability is undersampled by the ISAS13 strategy tested.

    The temperature data from one long model run is used for the analysis. The first 341 years of model output are dropped to

remove model spin-up effects, and the remaining 851 years of model output are separated into 37 23-year long segments; the same temporal span as the CORA database. The model has an overall drift of 0.27 ZJ/yr, which is captured by the linear trend of the decomposition $\frac{\Delta Q}{\Delta t}$. As a result different segments have different mean OHC, which is captured by $Q_M$. The largest source of variability in OHC is the seasonal cycle. Figure 3 shows time series of $Q - Q_M$ and $Q_A$ for 10 of the 23-year segments. The seasonal cycle and the anomalies in $Q$ are readily apparent.

**4.5   Application of the Observing Strategy to Model Data**

Synthetic observations are taken from each model segment by linearly interpolating the model data in latitude, longitude and time to the latitudes, longitudes, and times of the observations in the CORA dataset, and using the nearest-neighbor model





grid cell in depth.[2] We thus use the locations of the actual drifters to make synthetic measurements, which differs from the technique used by Kamenkovich et al. (2009), where synthetic drifters were advected by the model flow.

To apply the ISAS-V6 OA to the synthetic observations, the parameters of the objective analysis must first be defined. We choose to use the original model grid as the analysis grid to most easily compare the 'observed' estimates with the 'true' model values, as then the 'true' values do not require regridding. The 'true' climatologies and variances are calculated directly from the original model data to compare to the 'observed' values which assess how the sampling and gridding procedure of ISAS-V6 performs under these ideal circumstances. The decorrelation length and time scales used are the same as those used for ISAS13. Using these parameters, global fields of 'observed' model temperature are synthesized from which time series of $Q(t)$ are computed. Statistical comparisons between the 'observed' and 'true' time series of $Q(t)$ and its components are then performed.

## 5 Results

### 5.1 Trends, Seasonal Cycles, and Variability

We separately quantify the skill of the ISAS13 observing strategy in capturing the trend, seasonal cycle, and anomaly. We compute the distribution of the differences between the 'observed' and 'true' quantities and determined the mean and standard deviations of these distributions. The mean of the distribution quantifies the bias introduced by ISAS13, and the standard deviation of the distribution quantifies the expected error or uncertainty in ISAS13's estimation of these values.

The mean and standard deviation of the difference between the all-time 'observed' and 'true' trends is $-0.17 \pm 0.43$ ZJ yr$^{-1}$. The results are $-0.32 \pm 0.46$ ZJ yr$^{-1}$ for the pre-Argo era and $-0.17 \pm 0.40$ ZJ yr$^{-1}$ for the Argo era. The Argo era follows expectations and has a lower bias than the pre-Argo era. However, all trends and trend biases have one-sigma spreads greater than the mean. Since the mean of the trends and trend biases are both statistically indistinguishable from zero, it is impossible to determine whether the small bias is an actual bias in the observing strategy or just due to sampling variability about the small 'true' trend, which is solely due to model drift.

For the seasonal cycle we consider the monthly means of each segment and the amplitude of the seasonal cycle in the segment. The amplitude is here defined as half the range between the maximum and minimum monthly means. Averaging the monthly means over all months gives zero due to the definition of the decomposition. However, there are differences between the 'observed' and 'true' monthly mean for a given month and a given segment. Averaged over all months and all segments the rms difference between 'observed' and 'true' monthly means is 1.37 ZJ. The mean and standard deviation of the differences between the 'observed' and 'true' seasonal cycle amplitudes is $-1.04 \pm 0.17$ ZJ. This shows that the observing strategy underestimates the seasonal cycle amplitude by 2.7% relative to the 'true' amplitude of 37 ZJ, and that the variability in the bias across segments is small compared to the already small bias.

---

[2]No meaningful differences in results occur if nearest-neighbor sampling is used horizontally and in time instead of interpolation.





The amplitude of the natural variability in each segment is given by the rms value of $Q_A$. The mean and standard deviation of these amplitudes over the 37 'observed' segments is $4.0 \pm 2.3$ ZJ, which is slightly greater than that of the 'true' data, $3.6 \pm 2.5$ ZJ. For the pre-Argo era, the 'observed' and 'true' values are $2.9 \pm 1.7$ ZJ and $2.4 \pm 1.7$; for the Argo era, the values are $2.8 \pm 2.3$ and $2.9 \pm 2.3$, respectively. Thus, observations in the Argo era of observations do not, on average, inflate the amplitude of the natural variability.

The error in the anomaly is the difference between the 'observed' and 'true' values of $Q_A(t)$. Because the anomalies are defined to have zero mean, there is no mean bias. Averaged over all months and model segments, the rms of the errors in the anomalies is 3.5 ZJ for all-time, 3.3 ZJ for the pre-Argo era, and 1.4 ZJ for Argo era. The spread in the error in $Q_A$ introduced by the observing strategy is thus on the order of the 'true' $Q_A$ variability for the all-time and pre-Argo eras, but is only about half of the 'true' variability for the Argo era. This indicates that the ISAS13 observing strategy introduces significant error in the monthly variability of the global ocean heat content anomaly.

## 5.2 Timescale of errors due to sampling variability and OA: cross-correlations

We now ask whether the error in observing the OHCA can be constrained to a specific time scale, in which case temporal smoothing over this time scale will effectively remove the errors from the 'observed' estimates of global ocean heat content variability. Effects due to the location and timing of observations and due to the OA are the same for synthetic observations taken at the same synthetic time across different independent ensemble members. This introduces a correlation between different 'observed' segments. The anomaly of independent model runs would, on the other hand, be uncorrelated at the same synthetic times. Thus, these effects can be detected by looking at the cross-correlation of the ensemble members. The 'true' segments are not strictly independent since they are created by partitioning a single long integration. The time-lagged correlation of the 'true' $Q_A$ over the entire model run (apart from the spin-up) shows a correlation time of about 18 years. Cross-correlations of 'true' segments would, however, not be zero even if the segments were truly independent since they are finite in length. The cross-correlations of temporally adjacent segments shows a small positive bias, while cross-correlations of segments separated by one intervening segment (23 years) are indistinguishable from those of segments separated by two or more intervening segments. We thus only use cross-correlations of temporally non-adjacent segments in what follows. The distribution of cross-correlations of 'true' segments quantifies the spurious correlations due to finite size time segments. A difference in the distribution of cross-correlations of 'observed' segments from that of 'true' segments detects contamination of the OHCA by the observing system.

We now partition $Q_A$ into low and high frequency components with a cutoff time $\tau$ and investigate how the cross-correlations depend on the cutoff $\tau$. The goal is to find the minimum cutoff time for which the pdf of cross-correlations between 'observed' segments of $Q_L$ is roughly indistinguishable from that of 'true' segments, thus effectively confining spurious correlations introduced by the observing system to the high frequency component. The left column of Figure 4 shows that for $Q_L(t; \tau)$, the pre-Argo era (panel a) exhibits significant non-zero cross-correlations for subannual variability ($\tau < 12$ months), which also shows up for the all-time results (panel e). This indicates that observing strategy introduces spurious correlations on subannual



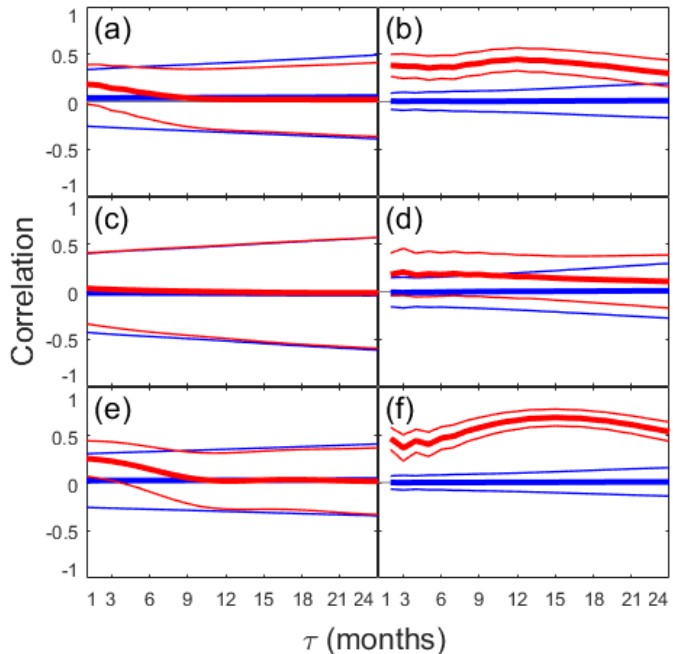

**Figure 4.** Means (thick) and standard deviations (thin) of the distributions of cross-correlations between the 630 'true' segments (blue) and 'observed' segments (red) for the low frequency anomaly $Q_L(t; \tau)$ (left: a,c, e) and high frequency anomaly $Q_H(t; \tau)$ (right: b,d,f) for values of averaging cutoff $\tau$. Disagreement between the 'observed' and 'true' distributions implies spurious correlations due to the observing strategy. Cross-correlations are for the pre-Argo era (top: a,b), Argo era (middle: c, d), and all-time periods (bottom: e,f).

time scales. For the Argo Era, spurious correlations are much smaller at short cutoff times as seen in the high similarities between the 'true' and 'observed' distributions (panel c).

The right column of Figure 4 shows that for the high frequency anomaly $Q_H(t; \tau)$, spurious correlations are prominent for all time scales in the pre-Argo data (panel b), which in turn effects the results for the all-time results (panel f). The observed cross-correlations of $Q_H(t; \tau)$ peak at about $\tau = 12$ months, indicating that subannual time scales contain the bulk of the spurious correlations. For the Argo Era, the distributions begin to significantly overlap at about $\tau \approx 12$ months, indicating that subannual time scales, even in the Argo Era, contain spurious correlations. Therefore, it can be concluded that annual running means are satisfactory to remove the spurious correlations introduced by the ISAS13 observing strategy, at least in reproducing this model's variability, and that variability due to the observing strategy dominates 'observed' subannual variability for the pre-Argo era.

### 5.3 Skill of ISAS13 Observation Strategy for Natural Variability: Correlation between 'observations' and 'truth'

We now quantify the capability of the observing strategy to detect the 'true' variability. The correlations between corresponding pairs of 'true' and 'observed' segments are computed to quantify this skill. The closer the correlations are to unity, the greater





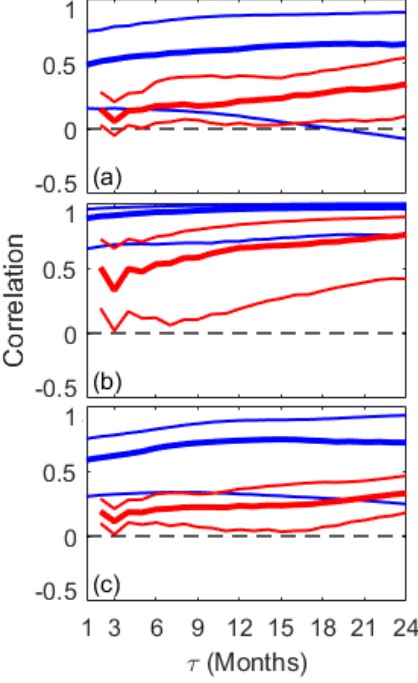

**Figure 5.** Distributions of the correlations between the 37 corresponding 'true' and 'observed' time series of $Q_L(t;\tau)$ (blue) and $Q_H(t;\tau)$ (red) as a function of filter cutoff time $\tau$ for the pre-Argo era (a), the Argo era (b), and all-time periods (c). The thick curves indicate the median of the distributions and the thin curves indicate the 5%-95% percentile range.

the detection skill of the observing strategy. The results are shown in Figure 5. Predictably, the low frequency variability $Q_L(t;\tau)$ is consistently better estimated than the high frequency variability $Q_H(t;\tau)$. For the Argo Era, the median correlation of $Q_L$ exceeds 0.95 for $\tau \geq 11$ months. The pre-Argo and all-time results are similar, and the median correlations of $Q_L$ never go beyond 0.75 for any cutoff time out to 90 months. Even in the Argo era, where the median of the distribution of the correlations of between 'observed' and 'true' low frequency interannual anomalies is 0.95, the full distribution is broad, reaching down to 0.71. This indicates that while, on average, the low frequency interannual anomaly is well captured by the Argo-era observing strategy, there are ensemble members that are not captured well, indicating that the ability to capture the low frequency anomaly depends on the specific state of the ocean. This is especially true in the pre-Argo era, where the spread in the 37 correlations spans from 0 to 0.75 once $\tau$ reaches 18 months.

## 5.4 Error Quantification

Now that annual running means have been identified as being satisfactory in removing artifacts due to the observing strategy and better representing true ocean variability than the original monthly time series, the ISAS13 observing strategy is examined with annual filtering. Using $Q_L(t)$ defined by (5) with $\tau = 12$ months, the distribution of the differences between the 37



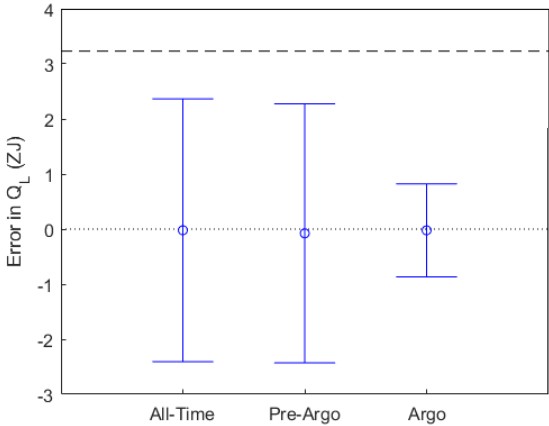

**Figure 6.** Mean and standard deviation of the errors between the 'observed' $Q_L(t; \tau = 12\text{months})$ and 'truth' averaged over all segments for the entire 23 year period of the segment, the pre-Argo era, and the Argo era. The dashed line is the root-mean-square value of the 'true' $Q_L$ over all segments. The signal-to-noise ratios of the standard deviation of the errors to the 'true' variability are 1.8 for all-time, 1.9 for pre-Argo, and 14.7 for the Argo era.

corresponding 'observed' and 'true' segments can be used to estimate the mean and spread of the error introduced by the observing strategy.

The mean and standard deviation of the errors in $Q_L$ for all-time, pre-Argo, and Argo eras are shown in Figure 6, along with the 'true' variability amplitude of $Q_L$. The mean error in all periods is small, indicating that the observing strategy does not

bias the low frequency (i.e., interannual) variability. The typical size of the errors in the Argo era is roughly 1/3 of the size of the errors in the pre-Argo era. Yet even in the Argo era, the typical error size is a significant fraction of the amplitude of the low frequency anomaly. In other terms, the signal-to-noise ratio of the three standard deviations of error are 1.8 for all-time, 1.9 for pre-Argo, and 14.7 for the Argo era.

### 5.5    Implications for ISAS13 ocean observations

So far we have quantified the skill of the observing strategy using synthetic observations of model runs. We now use these results to estimate uncertainties in ISAS13 ocean observations. Since we now know that high frequency anomalies are contaminated by the observing strategy we estimate the errors on the annual running mean of the global ocean heat content observations $\bar{Q}$:

$$\bar{Q} = Q_M + t\frac{\Delta Q}{\Delta t} + Q_L, \tag{6}$$

where $Q_L$ is now the low-frequency anomaly with a filter cutoff of 12 months. The annual average removes the high frequency anomaly and the seasonal cycle.





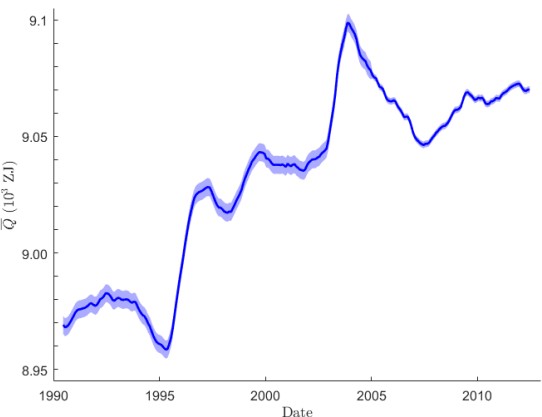

**Figure 7.** Annual running mean of the ISAS13 observed global ocean heat content and one-sigma uncertainty estimated by the EOSSE.

The errors in observing the trend were found in Section 5.1 above by applying the ISAS13 observing strategy to the model. We assume these errors represent the errors in the trend of the ISAS13 ocean observations as well. The ISAS13 data shows that the natural variability of the ocean is about 2.5 times larger than that of the model. We expect that the errors in observing the anomaly scale with the size of the anomalies. Our best estimates of the sizes of the anomalies are to use Argo-era 'true' model anomalies, which have an rms magnitude of 3.6 ZJ, and Argo-era ISAS13 ocean anomlies, which have an rms magnitude of 9.5 ZJ; their ratio, which is the uncertainty inflation factor is 2.6. We thus scale the uncertainties found in Section 5.4 by this uncertainty inflation factor and estimate the uncertainties separately for the pre-Argo and Argo eras. The resulting low-frequency observed global OHC along with its one-sigma estimated uncertainties is shown in Figure 7. One sees that after smoothing with an annual running mean, the uncertainties introduced by the observing system are small compared to the low-frequency variability and trend. We note that there may be uncorrected biases between pre-Argo and Argo instruments which produce the jumps seen at 1997 and 2003 which are an additional source of uncertainty.

Finally, we use the model results to estimate the uncertainty in the observed ISAS13 global ocean heat content warming trend. We take the anomalies of the model segments and multiply them by the uncertainty inflation factor defined above. These segments now have a variability magnitude similar to that seen in the real ocean. The trend of the segments is still extremely small, due to model drift. We now calculate the trend in the inflated 'observations' and measure the distribution of errors from the 'true' trend. This is done for linear trends beginning in each year from 1990-2011, and ending in 2012. From these distributions, half of the range covered by the 5% and 95% percentiles is used to estimate the uncertainty of the ISAS-13 observed ocean trend. The trend uncertainty is large for the 2-year period 2011-2012, 4.7 ZJ/yr, drops rapidly as the time span increases to five years, 2008-2012, to 1.7 ZJ/yr, and then slowly decreases until the uncertainty in the all-time trend, from 1990-2012, is 1.0 ZJ/yr. The 1990-2012 trend in the ISAS13 ocean data is observed to be 5.3 ZJ/yr. Thus we can conclude that errors in the observing strategy do not significantly contaminate the 1990-2012 warming trend.



## 6 Conclusions

We develop an Ensemble Observing System Simulation Experiment (EOSSE) procedure to quantify the skill of an observing strategy in capturing natural variability. The EOSSE method allows estimation of the distribution of errors, and the independent ensemble members allow a quantification of spurious correlations introduced by the observing strategy. We apply the method

to the ISAS13 observing strategy for global ocean heat content from 1990-2013 to the data from CCSM3.5 coupled model simulations. In this evaluation, subannual variability has significant spurious correlations introduced by the observing strategy. The strategy's most skillful estimates are found for annual running means from 2005 onwards, when the Argo program dominates the observing record. For the record of pre-Argo observations, the standard deviation of the errors in the annual running mean of global ocean heat content anomalies was found to be 2.4 ZJ, compared to only 0.8 ZJ for the record of Argo era observations.

The signal-to-noise ratio relative to the 'true' annual running mean variability of the ocean heat content anomalies is 1.9 for the pre-Argo record and 14.7 for the Argo record, indicating the Argo record increases the skill by an order of magnitude. Scaling the error in the model to account for the larger variability in the ocean, we estimate root-mean-square errors in observations of the ocean's annual running mean global ocean heat content anomaly of 6.2 ZJ in the pre-Argo era and and 2.1 ZJ in the Argo era.

The errors we quantify here are those introduced by measuring the ocean at the specific times and places of the historical record, and then using the ISAS13 objective analysis procedure to estimate global ocean heat content. Depending on the actual state of the ocean, represented by different model ensemble members, each global estimate differs from the 'true' value. Previous studies, which did not calculate distributions of errors, used a variety of techniques to estimate errors in the annual mean OHCA. All are within an order of magnitude of our results and each other (e.g. Kamenkovich et al. (2009), Zhang et al.

(2009), von Schuckmann and Le Traon (2011), Boyer et al. (2016)). Given that our error distributions are broad, that previous works are sampling from a single model run, and that important details differ across investigations, this spread in reported errors is not surprising.

We also examined the skill and uncertainty of the ISAS13 observing strategy in measuring the OHC trend. The uncertainty in the trend is 1.7 ZJ/yr for the five-year period 2008-2012, which then gradually decreases to a 1.0 ZJ/yr for the 1990-2012

trend. Relative to the observed trend in the ISAS13 product of 5.3 ZJ/yr for 1990-2012, we conclude that the errors in the ISAS13 observing strategy do not significantly contaminate the 1990-2012 warming trend. The ISAS13 trend and our derived uncertainty are similar to those found in other works (e.g., von Schuckmann and Le Traon (2011)'s 2005-2010 trend of $\sim 6 \pm 1$ ZJ/yr).

As with any OSSE technique, the estimates are restricted by the accuracy of the synthetic data. While this modeling system

has been shown to have skill in reproducing large-scale features and variability of the climate system, mesoscale and synoptic variability are underestimated or neglected. Furthermore, unlike the study of Kamenkovich et al. (2009) where Argo drifters are actually simulated as advective trajectories using the modeled data, here we preserve the locations and times of the actual observing system. Preserving these locations simplifies our ensemble experiment, but prevents a determination of the biases induced by trajectory-climate correlations.



Different mapping methods do differ in errors and estimates of global ocean heat content, but the broad similarity of the various methods leads us to speculate that our finding that subannual variability is significantly contaminated by the observing strategy is robust, and that only interannual and longer timescales of global ocean heat content variability and trends can be skillfully measured.

## 7 Code availability

The In-Situ Analysis System tool version 6 (ISAS-V6) objective analysis is a FORTRAN package maintained by Dr. Fabienne Gaillard of Ifremer. Please contact her at Fabienne.Gaillard@ifremer.fr for inquiries regarding the availability of the ISAS-V6 code.

## 8 Data availability

The rhlow4 simulation CCSM3.5 data analyzed here is available upon request to baylor@brown.edu, as are the other simulations described in Stevenson et al. (2012).

The Coriolis Ocean Dataset for Reanalysis (CORA) version 4.0 can be obtained via the following url hosted by SEANOE sea science open data publication: http://www.seanoe.org/data/00351/46219/

The ISAS13 dataset can be obtained via the following url hosted by SEANOE sea science open data publication: http://www.seanoe.org/data/00348/45945/

*Competing interests.* The authors declare they have no conflict of interest.

*Acknowledgements.* This work was funded by NSF INSPIRE Award #1245944. We would like to acknowledge high-performance computing support from Yellowstone (ark:/85065/d7wd3xhc) provided by NCAR's Computational and Information Systems Laboratory, sponsored by the National Science Foundation. We also thank the members of Ifremer responsible for maintaining the CORA data set for allowing us to use their optimal interpolation procedure, aiding in its implementation, and understanding the results. We also thank Dibyendu Mandal and the research groups of Weiqing Han and Baylor Fox-Kemper for participating in helpful discussions.



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
