# Peer review of "An Ensemble Observing System Simulation Experiment of Global Ocean Heat Content Variability"

_Ocean Science, 2016_

## Author Comment (AC1) · 18 Jan 2017

In the legend of Figure 1, the colors for the first 2 entries(Profiling Floats and XBTs, etc.) should be switched such that the blue curve corresponds to Profiling Floats and the orange curve corresponds to XBTs, etc.

In Figure 2, the caption does not provide adequate description for the 2 panels. Here's an updated caption: The locations of retained hydrographic profiles in the pre-Argo era of 1990-2005 (a) and the Argo era of 2005-2013 (b). Before Argo, observations were sparse and primarily by XBTs. Poor record keeping requires many of these observations to be from 'unknown' sources. The majority of the unsampled regions in the Argo era occur in regions shallower than 700m, although some are geopolitically sensitive areas.

Regards, Arin Nelson

---

## Referee Comment (RC1) · Anonymous Referee #1 · 22 Jan 2017

Nelson et al. quantified skill and uncertainty in observing the ocean heat content (OHC) in the upper 700-m layer using observing system simulation experiments (OSSEs). This is a highly recommended method to evaluate both historical observation system and the current mapping strategies in OHC studies, so the topic is very interesting and useful. However, when I read this paper through and observe the conclusion of this study, major deficiencies are found. The conclusion is rather shallow: the authors find that the uncertainty of OHC estimate pre-Argo era is larger than the Argo era. This is a broadly known issue, so it is not novel contribution. And the uncertainties values quoted in this study are not reliable, because this is a model-based study and the uncertainty depends largely on the model performance and model resolution (see my comments below). Besides, there are some other major flaws in this study, so I recommend rejection.

**Major**

1. The model resolution is 340km at the equator and 40km near the north polar. This is a low-resolution model, which means the model mainly simulates large-scale ocean variabilities, the smaller-scale variabilities such as meso-scale eddies are not resolved. This is a major difference between model and observation, since observation contains variations at all scales. Therefore, using models will significantly under-estimate the uncertainties. This main deficiency makes the uncertainty values provided in this study useless. This is my first major concern.

2. Page 2, line1-4. The authors said the focus of this study is to quantify the reliability of the warming trend. But I don't think the paper did that. There are two key questions related to the observed OHC trend: (1). Is the observed trend biased or not? This is a really intriguing question. (2). What is the uncertainty of the calculated trend? This study provides some clues for the second question based on OSSE method, but understanding the real uncertainty in OHC estimate is difficult and is not (and can not) sufficiently done by this model-based study. So the current study contributes little to the community.

3. The section-2 is to review the literatures, but it is incomplete, lack of many recent literatures about OHC estimation based on observations.

4. And, the review of the literatures in the section-2 is chaos, mixing observation-based studies and OSSE-based analyses together. For instance, Lyman et al. 2006 and 2008 are not OSSE. And the paragraph (lines around 20, page 3) seems strange in the context and von Schuckmann and Le Traon (2011) is not OSSE as well. Page4, the 2rd and 3rd paragraph, after two paragraphs discussing OSSE, those two go back to observation-based analyses again. Though I agree von Schuckmann et al and Lyman et al dealt with the climatology, did this paper have any contributions to clarify how the choice of climatology impacts the OHC calculation??? I think it will be helpful if

the authors could deal with this issue in the future using OSSE analysis (similar to Good et al. 2015), but it is not in the current manuscript.

5. Page 4, line 16-17. What the authors mean by "statistical error propagation methods."?? I think the authors are not clear at all about what the referred literatures did by so-called "objective analysis". Almost each objective analysis method dealt with uncertainty or error in their analyses.

6. Figure 7 is a horrible figure. It does not make any sense that the OHC time series should be like that!! The large jump around 1995 and 2003 must be spurious, so it is meaningless to show such a figure. The related discussion makes no sense as well. Moreover, the error bar is too small to be believable, it makes no physical sense that the error bar is so small. And also, why not provide OHC anomaly rather than OHC?

7. I suggest the authors give a clear definition about 'sampling strategy', 'observing strategy' and 'mapping method'.

8. Figure 1 is another horrible figure. I doubt the authors make it right. The profiling floats in Fig.1 are more than 90%, but it is not!!!!! See the figure below from NCEI. And the moored buoy should be much more than this figure shown. So it is not a trustable plot for me.

[Figure]

9.  Figure 2 is useless. I don't see the point of figure 2. It seems to me Fig.2a has very good global data coverage..

10. Another major confusion is about section 5.2. This section and related experiment is to investigate the dependence of timescale of errors due to sampling. However, the major variabilities of the model runs are on inter-annual scales (shown in Fig.3): i.e. there are no meso-scale signals, and no long-term trends. So it doesn't make any significant difference when using different window sizes ranging from 1 to 24 months. The small changes may reveal the uncertainty due to the small fluctuations added to the large inter-annual variation. I don't find it has any implications for long-term trend.

11. And what is the physical meaning of the standard deviation of the cross-correlation in Fig.4 and 5. If +/- one standard deviation means >60% confidence interval. The time variation of the mean correlation is not significant.

12. Similar to my point-10/11, in section 5.3 and Fig.3, the change of the correlation is neither significant nor physically tenable.

13. Section 5.4 is the only section that makes some senses, but the near-zero mean error in Fig.6 is not a surprise, since the models are free-runs without any external forcing. The ensemble mean should be zero. The only meaningful conclusion is the uncertainty pre-Argo is larger than Argo, but it is not surprise.

14. The authors argues that the size of the error in Argo era is 1/3 smaller than pre-Argo, I don't find it is a useful value. Because of many reasons (1). The model simulations in this study are mainly on inter-annual scales, no (weak) other variabilities, no trend etc. (2). The results should be specific for the ISAS mapping method and do not take account of any other errors (e.g. XBT bias, climatology issues)

---

## Author Comment (AC2) · 1 Feb 2017

This document is the author's response to review os-2016-105-RC1.

First off, we apologize that this reviewer expected this paper to be about the "reliability" of the long-term trend in ocean heat content. We explicitly state that we quantify the reliability of variability **about** the trend and seasonal cycle--i.e., not the trend, but the variations after detrending. There is extensive literature on the reliability of the observed warming of ocean heat content (e.g., *Domingues et al. 2008, Levitus et al. 2009*). Despite our efforts to clarify, there has been repeated confusion about the distinction between our study on the ocean heat content variability versus other studies on the ocean heat content trend.

As such, near the lines indicated by the reviewer as suggesting otherwise, we will add the additional clarifying statement:

"The focus of our study is *not* the reliability of the Ocean Heat Content trend over the whole record, which has been studied exhaustively elsewhere, but the reliability of variability once the trend has been removed--i.e., variability on timescales shorter than the record length."

Since the reviewer's conception of the paper was that it addressed the reliability of the observed trend, rather than of the variability of the observed trend, we found it difficult to apply all of the comments given by the reviewer. Nonetheless, the enumerated points by this reviewer are addressed below, with the reviewer's comments bolded.

1.
**The model resolution is 340km at the equator and 40km near the north polar. This is a low-resolution model, which means the model mainly simulates large-scale ocean variabilities, the smaller-scale variabilities such as meso-scale eddies are not resolved. This is a major difference between modeland observation, since observation contains variations at all scales. Therefore, using models will significantly under-estimate the uncertainties.**
We discuss this issue in Section 4.4, page 9, lines 14-19. We would love to have a high-resolution model output spanning thousands of years. Unfortunately, such models are too expensive computationally to be available this decade. In light of this, we used trusted model output that was already shown in previous studies to do a good job representing larger scale variability. This model was immediately available for our use. In other words, our choice of model was one of carefully-considered convenience.

**This main deficiency makes the uncertainty values provided in this study useless.**
We disagree, as the cited papers make it clear that the model possesses subannual variability consistent with observations (see Section 4.4, page 9, lines 12-14).

**2.**
**Page 2, line1-4. The authors said the focus of this study is to quantify the reliability of the warming trend.**
This is incorrect, as already mentioned. As is written on page 2, line 2-4; "Our focus here is to quantify the ability to faithfully measure the variability about the warming trend using an (EOSSE)".

**There are two key questions related to the observed OHC trend: (1). Is the observed trend biased or not? This is a really intriguing question. (2). What is the uncertainty of the calculated trend?**
As stated previously, the focus of this paper is not on the OHC trend, but on OHC anomaly variability, where we define 'anomaly' in this context to be about the all-time trend and seasonal cycle. Bias implies that the trend is either too fast or too slow, rather than variable in both directions on shorter timescales than the whole record, which is our focus.

**3.**
**The section-2 is to review the literatures, but it is incomplete, lack of many recent literatures about OHC estimation based on observations.**
We kindly ask the reviewer to provide references to some of the "many recent literatures" they mention here. We encourage others as well to suggest papers that may be beneficial to include in this section. We have read widely, and found the cited references to be of most direct relevance. This is, after all, not a review paper.

**4.**
**And, the review of the literatures in the section-2 is chaos, ...**
We attempted to reference previous estimates of global OHC uncertainty, not just from OSSEs but from other sources as well, such that we could compare our uncertainties to the uncertainties due to other factors in ocean observation processing. We apologize that the reviewer found this presentation to be disorganized.

**5.**
**Page 4, line 16-17. What the authors mean by "statistical error propagation methods."?? I think the authors are not clear at all about what the referred literatures did by so-called "objective analysis". Almost each objective analysis method dealt with uncertainty or error in their analyses.**
In the context of our referenced papers, the error estimates were either provided by the objective analysis method or based on similar mathematics. These methods are detailed exhaustively in the cited references (e.g., *Kaplan et al. 2000*), as well as exemplified in standard data analysis textbooks (e.g., *Emery & Thomson, 2001*). Our "errors" are constructed as differences from the given observations to the a-priori assumed climatological means and variances weighted by the number of observations within each specified region. Our method of uncertainty estimation does not depend on such a-priori assumptions.

**6.**

**Figure 7 is a horrible figure. It does not make any sense that the OHC time series should be like that!! The large jump around 1995 and 2003 must be spurious, so it is meaningless to show such a figure.**

We apologize that the large jumps in global OHC around 1995 and 2003 were not discussed in this paper. We have edited the caption of the figure, to reflect our conversations with Dr. Tanguy Szekely, the scientist in charge of the CORA dataset, copied verbatim below:

"The 1996 and 2003 jumps are associated to changes in the ocean sampling (see the PCTVAR field). The Indian ocean measurements almost disappear in 1996 leading to a decrease of the solution accuracy. In 2003, the begin of the worldwide ARGO deployment change the solution in numerous badly sampled zones (pacific ocean, South Atlantic Ocean, Antarctic Ocean, etc....) This problem will be solved in the next version of CORA (to be released on April 2016° A paper with details on the dataset description and validation should been published then."

Perhaps using the updated data set would 'fix' these jumps, but we are not addressing here how 'correct' the OHC time series is. The goal of this paper is to quantify the uncertainty due to the ISAS13 observing strategy, or construct the one-sigma confidence interval shown in the figure. The reason for including this figure was to compare the relative size of the confidence interval we find to features of the data, such as these jumps.

**The related discussion makes no sense as well.**

Without more specificity, we are unable to address this point.

**Moreover, the error bar is too small to be believable, it makes no physical sense that the error bar is so small.**

It must be emphasized here that the error bar is purely derived from the expected difference between the 'observed' and 'true' oceans from usage of the ISAS13 observing strategy. It does not take into account instrumental errors and biases, errors intrinsic to the objective analysis, mesoscale eddy activity, and other sources of error. It is clear from this comment that the reviewer has lost track of our argument by this point, for which we apologize.

**And also, why not provide OHC anomaly rather than OHC?**

Generally, OHC is considered always to be an anomaly as it has no meaningful baseline in a context such as this one; a zero-value of OHC has no dynamical or thermodynamical meaning. In the context of our paper, we define the OHC anomaly to be the OHC signal about the overall trend and seasonal cycle. To help clarify this point, Fig. 7 will be changed such that the all-time mean is removed.

**7.**

**I suggest the authors give a clear definition about 'sampling strategy', 'observing strategy' and 'mapping method'.**

An observing strategy in the context of our work contains two steps; the sampling strategy and the mapping method. The sampling strategy defines the locations and times of the observing strategy's set of observations. The mapping method describes the methodology used to process the resulting observations from the sampling strategy into "maps" with standardized locations and times (e.g., 1x1 degree grid of monthly means). We describe this in section 4.2, but will revise the discussion to clarify further.

**8.**

**Figure 1 is another horrible figure. I doubt the authors make it right. The profiling floats in Fig.1 are more than 90%, but it is not!!!!! See the figure below from NCEI. And the moored buoy should be much more than this figure shown. So it is not a trustable plot for me.**

In Figure 1, it is stated that our source of observations is the ISAS13 data collection. The ISAS13 dataset and the NCEI dataset do not contain the same observations, although there is some overlap. We invite others to reconstruct this figure and double-check our Figure 1.

**9.**

**Figure 2 is useless. I don't see the point of figure 2. It seems to me Fig.2a has very good global data coverage.**

This statement is subjective, as is the definition of "global coverage", but we understand that our small plot size in combination with the relatively large data points could incite this opinion. However, redrawing these figures to be larger with smaller data points would merely dwell further on subjectives and aesthetics. Many other references attempt a rigorous, objective definition of "global coverage" by ocean observations; it is not our intent to estimate such a quantity or precisely compute "global coverage" based on this figure, only to illustrate that the observations in the Argo era sample much more of the global ocean much more evenly than the pre-Argo observations.

**10.**

**Another major confusion is about section 5.2. This section and related experiment is to investigate the dependence of timescale of errors due to sampling. However, the major variabilities of the model runs are on inter-annual scales (shown in Fig.3): i.e. there are no meso-scale signals, and no long-term trends.**

This reviewer is again misled by his preconception that the focus of the paper is on the reliability of the trend. The model is run to near-equilibrium, so there is no long-term trend. We agree that mesoscale signals would add variability, which is impossible to assess based on our chosen model, but the models have significant annual and shorter timescale variability as it is a fully coupled model with synoptic winds, seasons, etc.

**I don't find it has any implications for long-term trend.**
Once again, the long-term trend is not our intended objective.

**11.**
**And what is the physical meaning of the standard deviation of the cross-correlation in Fig.4 and 5.**
We understand the confusion for the physical meaning of the standard deviations of the cross-correlations--this was not an obvious metric to us either when we began the project. Since the ISAS13 observing strategy only captures 23 years of the ocean, and we know the ocean has a great amount of variability on time scales longer than 23 years, the standard deviation of the cross-correlation distributions represents the sensitivity of the cross-correlations to the state of the ocean in the 23-year span of the observing strategy.  This definition follows for the pre-Argo era and Argo era of observations as well.

**If +/- one standard deviation means >60% confidence interval. The time variation of the mean correlation is not significant.**
This statement is incorrect.  In Figures 4(b,e,f), the distributions have statistically-significant changes in their means versus the time-window selection, at least to one-sigma (~68%) confidence.  For Figures 5(a,b,c), the change of the 50% percentile against the 5% to 95% percentile range versus time-window is unclear without performing some significance test. However, there is a statistically significant difference between the pre-Argo era and Argo era, which is the point this Figure aims to show.

**12.**
**Similar to my point-10/11, in section 5.3 and Fig.3, the change of the correlation is neither significant nor physically tenable.**
The reviewer must clarify how this point is differs from the ones made in 11.

**13.**
**Section 5.4 is the only section that makes some senses, but the near-zero**
**mean error in Fig.6 is not a surprise, since the models are free-runs without**
**any external forcing. The ensemble mean should be zero.**
This is true, the OHC anomaly time series have zero-means by construction, although why the reviewer pointed this out is unclear.  The important information contained in Figure 6 is the size of the standard deviation of the errors relative to the variability in the data represented by the dashed line.  This gives information about the signal-to-noise ratio, an extremely important statistic in signal processing, and a primary result of this work.

**The only meaningful conclusion is the uncertainty pre-Argo is larger than Argo, but it is not surprise.**

It is true that this result is no surprise. However, the difference in the uncertainty in the pre-Argo versus the Argo era is rarely quantified with any precision. Some variability in OHC would be detectable even in the pre-Argo era, and some would not. Figure 6 indicates quantitatively where the boundary lies in this particular model. One of the goals of this paper was to introduce a methodology that can quantify such quantities, which we feel we've demonstrated successfully. Such a quantification is critical in determining the accuracy of reconstructions of historical variability.

**14.**

**The authors argues that the size of the error in Argo era is 1/3 smaller than pre-Argo, I don't find it is a useful value. Because of many reasons (1). The model simulations in this study are mainly on inter-annual scales, no (weak) other variabilities, no trend etc.**

See our response to point 10. Furthermore, we are repeatedly explicit that this paper presents a methodology for evaluation of these statistics which might be used with any model. Other models would have different variability which would need the same kind of quantification.

**(2). The results should be specific for the ISAS mapping method and do not take account of any other errors (e.g. XBT bias, climatology issues)**

This is true. The methodologies we introduced are to objectively quantify the errors associated with a specific observing strategy (mapping and sampling pattern) and do not take into account errors due to instrumental biases and errors, climatology choice, etc. One of the goals of this work was to quantify this specific source of uncertainty. Other sources of uncertainty have been studied in other papers, but we believe attention to how well variability is observed is a novel question.

**Bibliography**

[1] Domingues et al. 2008, Improved estimates of upper-ocean warming and multi-decadal sea-level rise, *Nature* **453**, 1090-1093.

[2] Levitus et al. 2009, Global ocean heat content 1955-2009 in light of recently revealed instrumentation problems, *Geophysical Research Letters* **36**(7).

[3] Kaplan, A., Kushnir, Y. & Cane, M. A. Reduced space optimal interpolation of historical marine sea level pressure. J. Clim. 13, 2987–3002 (2000).

[4] William J. Emery and Richard E. Thomson, Data Analysis Methods in Physical Oceanography, Elsevier Science, Amsterdam, 2001, ISBN 9780444507563.

---

## Referee Comment (RC2) · Anonymous Referee #2 · 28 Feb 2017

This work uses the Community Climate System Model (CCSM) 3.5 to investigate uncertainty in variability of ocean heat content (OHC) due to observation distribution for the years 1990-2013. The paper breaks down OHC into a mean, seasonal cycle, secular trend, and anomaly. The paper further breaks down the anomaly into high and low frequency anomalies in time with the purpose of quantifying uncertainty in the OHC anomaly and setting a time period cutoff to minimize high frequency noise. The paper finds that the Argo period (2005-2013) coverage leads to lower uncertainty than the pre-Argo period (1990-2004) and that yearly or longer time periods have sufficiently low signal-noise ratio to confidently estimate the OHC anomaly.

The paper is innovative in its approach breaking down the OHC and using a model run to investigate uncertainty in the anomaly, The investigation of uncertainty in the anomaly is thorough. The main conclusion of this exercise (that the Argo period has

lower uncertainty and that OHC estimates for the yearly or longer time period should be used due to higher uncertainty at higher time frequencies) are not groundbreaking. Most estimates of OHC change use yearly or longer time frequencies, and it is well documented that the Argo data coverage is better than previously. That is not to say that there is not merit in these conclusions from rigorous statistical analysis.

The authors do not look at uncertainty in the secular (non-cyclical) change from the model, rather looking at variability around a secular linear trend. The authors note (on page 2, top) 'the variability about the warming trend has yet to be reliably quantified due to the underlying uncertainties'. I am not sure the authors have done quite that here. In their breakdown of OHC, they represent secular change as a trend (deltaQ/deltat). Not much is said about how this is done, but secular change represented by a trend will unavoidably influence the OHC anomaly [if the trend is overestimated, anomaly will be biased negative, understimated, biased positive, etc]. The obvious solution would be to run the model without secular change - meaning without anthropogenic change. Then only cyclical natural (non-human induced) variability would contribute to the OHC anomaly. But the authors note that the model they use is 'run without variability in anthropogenic forcing'. Unless that constant forcing is zero, there will still be a secular change and a contamination of the OHC anomaly. The other conclusion, validating the ISAS13 OHC change 1990-2013 will be discussed in a following paragraph.

I also think the authors could use some better terminology in the paper. This is a mild suggestion only, the paper could be published with the current terminology, maybe with some more prominent definition. The authors use the word 'truth' throughout to mean results from the geographically complete model run as opposed to the 'observed' results from the geographically subsampled and objectively analyzed fields. They also occasionally refer to the 'real' ocean. Not being a modeler, it is somewhat incongruous to hear the model results referred to as 'truth'. Maybe 'model truth' or 'complete model' juxtaposed against 'subsampled', whereas the 'real' ocean would be the 'truth'. The authors define 'natural variability' as 'the variability of the ocean not forced by multidecadal or slower linear climate change and seasonality'. But there is no term for mult-decadal climate change in their OHC equation, except the anomaly term. It would make sense to equate more closely the term 'natural variability' to the OHC anomaly, as the terms are used interchangeably in the paper. The final term which could use amendment is 'ISAS13 observing strategy'. ISAS13 is a set of gridded fields of temperature and salinity. It does not have a set plan for sampling the ocean, an observing strategy. What ISAS13 has is an 'observed data distribution' from which it calculates t and s fields.

Section 5.5 of the paper is not easy to follow, but it provides the results which lead to the conclusion that ISAS13 OHC trend is not significantly contaminated. First, there is little in the way of description of the ISAS13 field calculation. The first time ISAS13 is mentioned there needs to be a reference. This reference should include details of the objective analysis procedure and full details of the OHC calculations - including climatology used and XBT bias correction applied. The authors do not consider in this section any source of uncertainty except the uncertainty in the OHC anomaly in the estimate of the linear trend of OHC change. The authors further note twice in the paper, including the concluding sentence, that varying mapping method (a term which should be defined) will not affect the results of the present work. But previous work has shown that the method used to extrapolate and smooth irregular data does have a large contribution to uncertainty. So, even if mapping method would not affect the results of this work (a speculation in the authors conclusion), it does have an impact on the uncertainty of the trend in OHC change. Likewise XBT bias correction. So saying that OHC anomaly uncertainty alone is low enough not to contaminate the calculated trend is not sufficient. Other sources of uncertainty should also be factored in, or if the OHC anonaly uncertainty is an uncertainty which incorporates these other uncertainties in some way, this should be explained in detail. I would like to see more explanation of the uncertainty inflation factor. Is this a standard practice? Why is it statistically valid here? The authors note that ISAS13 variability is about 2.5 that of the model variability - but the model was run with steady anthropogenic forces which I would expect to

dampen variability (at least secular variability). So the low variability may be due to the unrealistic forcing. Finally, figure 7 raises a number of questions, starting with the units. It is impossible (for me) to see the power of 10 for the y axis in the figure. This, and the caption and accompanying text noting '(low frequency) observed global ocean heat content' make it hard to know what exactly is being shown. Is this Qbar which includes mean OHC, a huge number in comparison with deltaQ/deltat and QsubL, or is this just the latter two terms? The uncertainty of the anomaly is necessarily very small compared to the entire global OHC (as the anomaly is just a fraction of the global OHC), shouldnt just the deltaQ/deltat and QsubL be shown with the QsubL uncertainty? I am also not sure how the jumps in year 1997 has anything to do with the change in observing system. There were no Argo floats in 1997, and only a handful of profiling floats. I would also like to see more explanation for the peak around 2004 and subsequent drop in relation to the change in observing system, since the observing system was already dominated by Argo in 2004. Can the authors please add more explanation to the text and figures for Section 5.5?

The paper is generally well written and clear (with the caveats noted above). There is an extra 'non' in 'non non-uniform' at line 6 on the first page. There is a missing 'a' in 'anomlies on line 5 of page 15.

Figure 1 has incorrect labeling. Blue line should be profiling floats. In addition the counts for other instrument types are too low. A quick look shows there were more than 80,000 XBTs dropped in 1990 for instance, an average of more than 6,000/month.

---

## Author Comment (AC3) · 21 Apr 2017

This document is the author's response to review os-2016-105-RC2.

We will address the comments of Anonymous Referee #2 in the following fashion:
- Referee's comment (bolded).
- Our response to the comment(s).
- Our change(s) to the manuscript.

**1.**
**The authors do not look at uncertainty in the secular (non-cyclical) change from the model, rather looking at variability around a secular linear trend. The authors note (on page 2, top) 'the variability about the warming trend has yet to be reliably quantified due to the underlying uncertainties'. I am not sure the authors have done quite that here. In their breakdown of OHC, they represent secular change as a trend (deltaQ/deltat). Not much is said about how this is done, but secular change represented by a trend will unavoidably influence the OHC anomaly [if the trend is overestimated, anomaly will be biased negative, understimated, biased positive, etc]. The obvious solution would be to run the model without secular change - meaning without anthropogenic change. Then only cyclical natural (non-human induced) variability would contribute to the OHC anomaly. But the authors note that the model they use is 'run without variability in anthropogenic forcing'. Unless that constant forcing is zero, there will still be a secular change and a contamination of the OHC anomaly.**

The referee's comments are correct, but are unfortunately misdirected by our awkward writing to have formed a misconception on the referee's part. We took 'anthropogenic forcing' to be related to *changes* in the atmospheric greenhouse gas concentration. Since the model was run without changes in greenhouse gas concentration, we took this to mean that there was no anthropogenic forcing in the model. This statement should be re-written as 'run with constant forcings', as there were no anthropogenic changes as well as no volcanoes, no changes in solar insolation over time, and other terms that could be considered 'variable' forcings. I.e., the greenhouse gas concentration is held fixed near the 1990 value. Note that even though the forcing is held constant, there may be some secular trend due to remaining model spin-up in some variables, thus a detrending is still required.

We will change the text on page 9, line 14, from (Orig.) to (Edit):
(Orig.) …, is run without variability in anthropogenic forcing, …
(Edit)   …, is run without variability in the external forcings, …

**2.**

**I also think the authors could use some better terminology in the paper. ... The authors use the word 'truth' throughout to mean results from the geographically complete model run as opposed to the 'observed' results from the geographically subsampled and objectively analyzed fields. They also occasionally refer to the 'real' ocean. Not being a modeler, it is somewhat incongruous to hear the model results referred to as 'truth'. Maybe 'model truth' or 'complete model' juxtaposed against 'subsampled', whereas the 'real' ocean would be the 'truth'.**

We understand the referee's concern here as we have struggled with the terminology ourselves. We agree with the referee that the 'model truth', 'observed model', and 'observed ocean' are more sensible than the terms we used; 'truth', 'observed' and 'real'.

We will change these terms throughout the manuscript.

**3.**

**The authors define 'natural variability' as 'the variability of the ocean not forced by multi-decadal or slower linear climate change and seasonality'. But there is no term for multi-decadal climate change in their OHC equation, except the anomaly term. It would make sense to equate more closely the term 'natural variability' to the OHC anomaly, as the terms are used interchangeably in the paper.**

The referee's comment is correct. We originally defined OHC anomaly to be equivalent to the natural variability and understand the confusion in wording the natural variability as 'the variability in the ocean not forced by multi-decadal or slower linear climate change in seasonality". Since the record length in our case is 23 years, multi-decadal variability would be difficult to distinguish, but in general this isn't necessarily true.

We will change the text on page 8, line 11, from (Orig.) to (Edit).
(Orig.) … to be the variability of the ocean not forced by multi-decadal or slower linear climate change and seasonality. …
(Edit.) … to be the variability of the ocean about the record-length seasonal cycle and linear trend. …

**4.**

**The final term which could use amendment is 'ISAS13 observing strategy'. ISAS13 is a set of gridded fields of temperature and salinity. It does not have a set plan for sampling the ocean, an observing strategy. What ISAS13 has is an 'observed data distribution' from which it calculates t and s fields.**

We understand the referee's concern for the term 'ISAS13 observing strategy'. We wish to express the combination of the observed data distribution and objective analysis methodology used in the creation of the ISAS13 data set in as few words as possible, and we originally used the term 'ISAS13 observing strategy' without a thorough discussion of its meaning. Therefore, we will add a descriptive definition of what we mean by the 'ISAS13 observing strategy' and by the term 'observing strategy' in a general sense in Section 4.2.

We will change Section 4.2 to read the following:

ISAS13 is a set of gridded global fields of surface and subsurface ocean temperature and salinity on a 0.5x0.5 degree grid spanning the years 1990 through 2012 (Gaillard et al. 2016). ISAS13 is constructed by gathering the observations from a variety of sources, including Argo, (etc.) and applying an objective analysis (OA) procedure to generate the gridded global fields from the point observations. The OA uses the optimal interpolation technique presented by Bretherton et al. (1976) and depends on a mean reference (or first guess) field, a field of the expected variance about the mean field, and de-correlation length and time scales. In the construction of the ISAS13 data set, this OA uses the monthly means provided by the World Ocean Atlas 2005 (WOA05) monthly climatologies (Locarnini et al. 2006, Antonov et al. 2006), monthly variances estimated from the observations relative to these climatologies, and decorrelation time scales chosen to represent 1) the resolution of the Argo network and 2) the first-mode Rossby radius of deformation on each point of their chosen analysis grid.

While a strategy assumes an a-priori plan, we define an 'observing strategy' as the methodology employed to produce a global subsurface ocean data set, whether the observational meta-data was planned before-hand or not. In our case, the ISAS13 observing strategy is defined as the set of observation locations and times plus the OA used to generate the ISAS13 data set, even though the observations and OA parameters were available beforehand.

We will also change the last paragraph of Section 1 (starting on line 21) to the following:

We demonstrate the method by quantifying the uncertainty associated with the spatio-temporal variability of ocean observations and specific choice of the mapping method used to construct global gridded fields from these observations associated with the ISAS13 data set (Gaillard et al. 2016). Specifically, the uncertainties are quantified for the estimate of global ocean heat content (OHC) variability down to 700m between 1990 and 2013. This 'observing strategy' is applied to 37 independent model segments from the Community Climate System Model version 3.5 (CCSM3.5), from which the statistics of OHC variability will be compared between the 'observed model' segments and their corresponding 'true model' segments across a range of time scales. The application of the methodology is described in Section 4, and the results are discussed in Section 5.

**5.**
**… There is little in the way of description of the ISAS13 field calculation. The first time ISAS13 is mentioned there needs to be a reference. This reference should include details of the objective analysis procedure and full details of the OHC calculations - including climatology used and XBT bias correction applied.**

These comments are addressed in the response to the previous comment.

**6.**
**The authors further note twice in the paper, including the concluding sentence, that varying mapping method (a term which should be defined) will not affect the results of the present work. But previous work has shown that the method used to extrapolate and smooth irregular data does have a large contribution to uncertainty. So, even if mapping method would not affect the results of this work (a speculation in the authors conclusion), it does have an impact on the uncertainty of the trend in OHC change. Likewise XBT bias correction. So saying that OHC anomaly uncertainty alone is low enough not to contaminate the calculated trend is not sufficient. Other sources of uncertainty should also be factored in, or if the OHC anomaly uncertainty is an uncertainty which incorporates these other uncertainties in some way, this should be explained in detail.**

Here is the concluding sentence of the submitted manuscript:

"Different mapping methods do differ in errors and estimates of global ocean heat content, but the broad similarity of the various methods leads us to speculate that our finding that subannual variability is significantly contaminated by the observing strategy is robust, and that only interannual and longer timescales of global ocean heat content variability and trends can be skillfully measured."

In this statement, we do not mean to state that varying the mapping method will not affect the results of our work. What we mean to say is that the spatio-temporal sampling dominates the error between the 'observed model' and 'true model' global OHC estimates in our study, and changing the mapping method won't change this conclusion. This conclusion comes from the large errors seen in the pre-Argo era versus the Argo era. Since the mapping method is the same for both eras, but the spatio-temporal sampling is very different in the two eras, we conclude that the effect of sampling is larger than the effect of the mapping method on the global OHC estimate. However, if the Argo era is examined alone, it is impossible to determine from this work alone how large the choice of mapping method will affect the estimates of global OHC variability. The manuscript will be re-worded to make this distinction clear. Additionally, since the observations are taken in a 'perfect-model' scenario, choice of XBT bias correction (and other related bias/error corrections) is not something that can be assessed in this methodology.

We will change the concluding paragraph of the manuscript quoted above to the following:
    From the large differences seen in the pre-Argo era relative to the Argo era, since the mapping method is the same for both eras, we find that the spatio-temporal variability in sampling is the largest source of error in the ISAS13 observing strategy. It is impossible to determine from this work alone how

large an effect the choice of a mapping method will have on these errors, but the similarity amongst the objective analyses used in the cited literature leads us to speculate that this result will remain unaffected by the choice of mapping method.  We therefore conclude that subannual variability is significantly contaminated by this observing strategy, and that only interannual and longer timescales of global ocean heat content variability and trends can be skillfully measured.  While other works have come up with this same conclusion, this work is perhaps the first to quantify this measurement skill, as given previously.

**7.**
**I would like to see more explanation of the uncertainty inflation factor. Is this a standard practice? Why is it statistically valid here? The authors note that ISAS13 variability is about 2.5 that of the model variability - but the model was run with steady anthropogenic forces which I would expect to dampen variability (at least secular variability). So the low variability may be due to the unrealistic forcing.**

We do expect the model to have unrealistic variability, and as stated in the second paragraph of Section 4.4, page 9, lines 12-19, "the estimates found using this model are based on a necessarily less variable climate system".  In order to increase ('inflate') the model variance to that of the real-world ocean, we multiplied our uncertainties by the ratio of the observed 'real world' variance to the 'observed model' variance.  This is not standard modeling practice, but without significant change to our experimental design (e.g., examining a higher resolution model and doing another comparison study), we wanted to suggest an estimate for the real world uncertainty.  We will edit the manuscript to make it more clear to the reader that this is done only as a rough estimate of this estimate.

We will create a paragraph break after the sentence ending with "...shown in Figure 7." in section 5.5 on line 3 of page 15.  The following sentences will be added to the end of this paragraph:
        It should be noted that this variance inflation method is not standard practice, but this method gives a rough estimate of real world uncertainty.

**8.**
**… Figure 7 raises a number of questions, starting with the units. It is impossible (for me) to see the power of 10 for the y axis in the figure. This, and the caption and accompanying text noting '(low frequency) observed global ocean heat content' make it hard to know what exactly is being shown. Is this Qbar which includes mean OHC, a huge number in comparison with deltaQ/deltat and QsubL, or is this just the latter two terms? The uncertainty of the anomaly is necessarily very small compared to the entire**

**global OHC (as the anomaly is just a fraction of the global OHC), shouldnt just the deltaQ/deltat and QsubL be shown with the QsubL uncertainty?**

The referee is correct, deltaQ/deltat and QsubL are much smaller than Qbar, and therefore it makes more sense to plot only QsubL + deltaQ/deltat * t to better visualize the magnitude of the temporal variability and uncertainties. This change will be made in the edited manuscript. The figure caption will also be changed to clarify what is being shown, per the referee's request.

We will change the caption of Figure 7 to the following:
(Orig.) Annual running mean of the ISAS13 observed global ocean heat content and one-sigma uncertainty estimated by the EOSSE.
(Edit) QsubL(tau=12) + deltaQ/deltat * t as calculated from the ISAS13 data set. The shaded region indicates the one-sigma uncertainty due to the ISAS13 observing strategy as estimated by the EOSSE methodology introduced in this work.

**9.**
**I am also not sure how the jumps in year 1997 has anything to do with the change in observing system. There were no Argo floats in 1997, and only a handful of profiling floats. I would also like to see more explanation for the peak around 2004 and subsequent drop in relation to the change in observing system, since the observing system was already dominated by Argo in 2004.**

We have emailed Dr. Tanguy Szekeley regarding this very concern. Here is the response we received addressing these jumps, copied here with some minor grammatical corrections:

The 1996 and 2003 jumps are associated to changes in the ocean sampling (see the PCTVAR field). The Indian ocean measurements almost disappear in 1996 leading to a decrease of the solution accuracy. In 2003, the begin of the worldwide ARGO deployment change the solution in numerous badly sampled zones (pacific ocean, South Atlantic Ocean, Antarctic Ocean, etc....)
This problem will be solved in the next version of CORA (to be released on April 2016, a paper with details on the dataset description and validation should been published then).

Due to recent release of the new data set, we were unable to re-run our analyses using the updated CORA data set (the computations were originally run in Fall 2015 through Winter 2016). We will update the manuscript to state this, as it is currently unknown whether the new CORA dataset does or does not contain these jumps at ~1996 and ~2003.

Together with the comments given in (**6.**), the last paragraph of Section 5.5, page 15, beginning on line 7, will be changed to the following:
        One sees that after smoothing with an annual running mean, the uncertainties introduced by the observing system are small compared to the low-frequency variability and trend. We note that there

may be uncorrected biases between pre-Argo and Argo instruments which produce the jumps seen at 1997 and 2003 which are an additional source of uncertainty. According to personal communications with Dr. Tanguy Szekeley, the curator of the CORA data set, these jumps and other problems have been resolved in the newest version of CORA referenced by Gaillard et al. (2016).

**10.**
**Can the authors please add more explanation to the text and figures for Section 5.5?**

This has been addressed in (**7.** - **9.**), but we have edited the text slightly in multiple places in Section 5.5, and have changed the caption of Figure 8 to the following:
(Orig.) Annual running mean of the ISAS13 observed global ocean heat content and one-sigma uncertainty estimated by the EOSSE.
(Edit) Annual running mean of the ISAS13 observed global ocean heat content (solid line) and its estimated one-sigma uncertainty bounds (shaded region). The uncertainties are estimated separately for the pre-Argo and Argo era's by taking the uncertainties derived from the EOSSE (see Figure 7) and multiplying them by the ratio of the variance in the ISAS13 Argo-era data to the variance of the full 'model truth'; 2.6.

**11.**
**There is an extra 'non' in 'non non-uniform' at line 6 on the first page. There is a missing 'a' in 'anomlies on line 5 of page 15.**

We thank the reviewer for pointing out these errors. They have been corrected in the updated manuscript.

**12.**
**Figure 1 has incorrect labeling. Blue line should be profiling floats. In addition the counts for other instrument types are too low. A quick look shows there were more than 80,000 XBTs dropped in 1990 for instance, an average of more than 6,000/month.**

This error was addressed in Author Comment #1 and will be implemented in the revised manuscript.

---

## Author Comment (AC4) · 14 Jun 2017

As pointed out by both reviewers, there appeared to be much fewer XBT profiles used in our analyses than expected. Following their logic, we found that there was indeed a shortcoming in our observational procedure. All casts that terminated shallower than 700m were rejected from our analysis, which was not the case in the ISAS13 data set. Therefore, we request a time extension to re-run our computations. The computations will take roughly 4-6 weeks, and then we will need time to revise our manuscript. We do not expect our conclusions to significantly change, but we do agree that the computations need to be redone. We therefore request an extension until August 18, 2017 to provide the time to make the necessary revisions.

[Figure]

We thank both reviewers for pointing out this error, and we will send the revised manuscript when this error has been rectified. We also thank you for your consideration in allowing us to fix our error before our manuscript is rejected. The revised manuscript will also include our responses to the previous rounds of reviewer comments, and so we choose to not respond to these comments at this time.

With thanks and warm regards, Arin Nelson and co-authors